# Realistic Evaluation of TabPFN v2.5 in Open Environments

**Zi-Jian Cheng** [* 1 2]  **Zi-Yi Jia** [* 1 2]  **Lan-Zhe Guo** [1 2]

## Abstract

Tabular data has garnered significant attention in machine learning research. While tree-based models have long dominated tabular machine learning tasks, the recently proposed deep learning model TabPFN v2.5 has emerged, demonstrating unparalleled performance and scalability potential. Although extensive research has been conducted on TabPFN v2.5 to further improve performance, the majority of this research remains confined to closed environments, neglecting the challenges that frequently arise in open environments. This naturally leads to an important question of **whether TabPFN v2.5 can maintain strong performance under open-environment challenges**. To this end, we conduct the first comprehensive evaluation of TabPFN v2.5's adaptability in open environments. We construct a unified evaluation framework covering various real-world challenges and assess the robustness of TabPFN v2.5 under this framework. Empirical results demonstrate that **TabPFN v2.5 shows significant limitations in data-fitting scenarios but is suitable for covariate-shifted, class-imbalanced, and prior-driven tasks**. To advance future research in open environments, we advocate for **open-environment tabular benchmarks featuring multi-metric evaluation, a strengthened emphasis on robustness, dedicated modules for TabPFN v2.5, and the use of synthetic data that reflects open-environment characteristics during training**.

## 1. Introduction

Tabular data is defined by its orthogonal row–column organization, where rows denote instances and columns denote

---
[*]Equal contribution [1]School of Intelligence Science and Technology, Nanjing University, China [2]National Key Laboratory for Novel Software Technology, Nanjing University, China. Correspondence to: Lan-Zhe Guo <guolz@nju.edu.cn>.

*Proceedings of the $2^{nd}$ ICML Workshop on Foundation Models for Structured Data*, Seoul, South Korea. 2026. Copyright 2026 by the author(s).

features (Altman & Krzywinski, 2017). The pervasive applicability of tabular data has been demonstrated across diverse domains (Zhu et al., 2021; Yıldız & Kalayci, 2024). To fully exploit tabular data for real-world tasks, various models have been developed, from tree-based methods (e.g., CatBoost (Prokhorenkova et al., 2018) and XGBoost (Chen & Guestrin, 2016)) to deep learning models (e.g., Modern-NCA (Ye et al., 2024) and TabPFN (Hollmann et al., 2023; 2025; Grinsztajn et al., 2025)). While tree-based models traditionally outperform deep learning on tabular tasks (Grinsztajn et al., 2022; McElfresh et al., 2023), TabPFN v2.5, a Transformer-based model (Vaswani et al., 2017) with large-scale pre-training on synthetic data, has disrupted this trend, achieving state-of-the-art performance across diverse tabular benchmarks (Grinsztajn et al., 2025).

Given the significant potential demonstrated by TabPFN v2.5 in handling tabular machine learning tasks, current research has focused on further enhancing its performance or adapting it to more real-world applications. However, these research on TabPFN v2.5 are mostly carried out in closed environments (Ye et al., 2025; Liu & Ye, 2025) where various learning factors, such as data distribution and feature space, remain consistent (Parmar et al., 2023). In the real world, tabular tasks usually occur in open environments (Zhou, 2022) and face significant challenges when these learning factors change. For example, in traffic management systems, as the categories of traffic participants, event types, and facilities continue to increase, the complexity of management significantly rises (**Emerging New Classes**). Meanwhile, equipment updates, failures, and changes in travel behaviour lead to feature drifts in data, affecting the system's accurate perception of traffic states (**Decremental/Incremental Features**). Moreover, the distribution of traffic flow frequently changes due to factors such as urban planning, large-scale events, and holidays, further increasing the dynamism of management (**Changing Data Distributions**). In addition, management goals have also shifted from single-efficiency optimization to multi-objective optimization, including reducing carbon emissions and enhancing system resilience, while paying more attention to long-term sustainability and overall system optimization (**Varied Learning Objectives**). Figure 1 depicts four open environment challenges claimed in (Zhou, 2022). Although existing research has gradually focused on improving TabPFN v2.5's adaptability in open

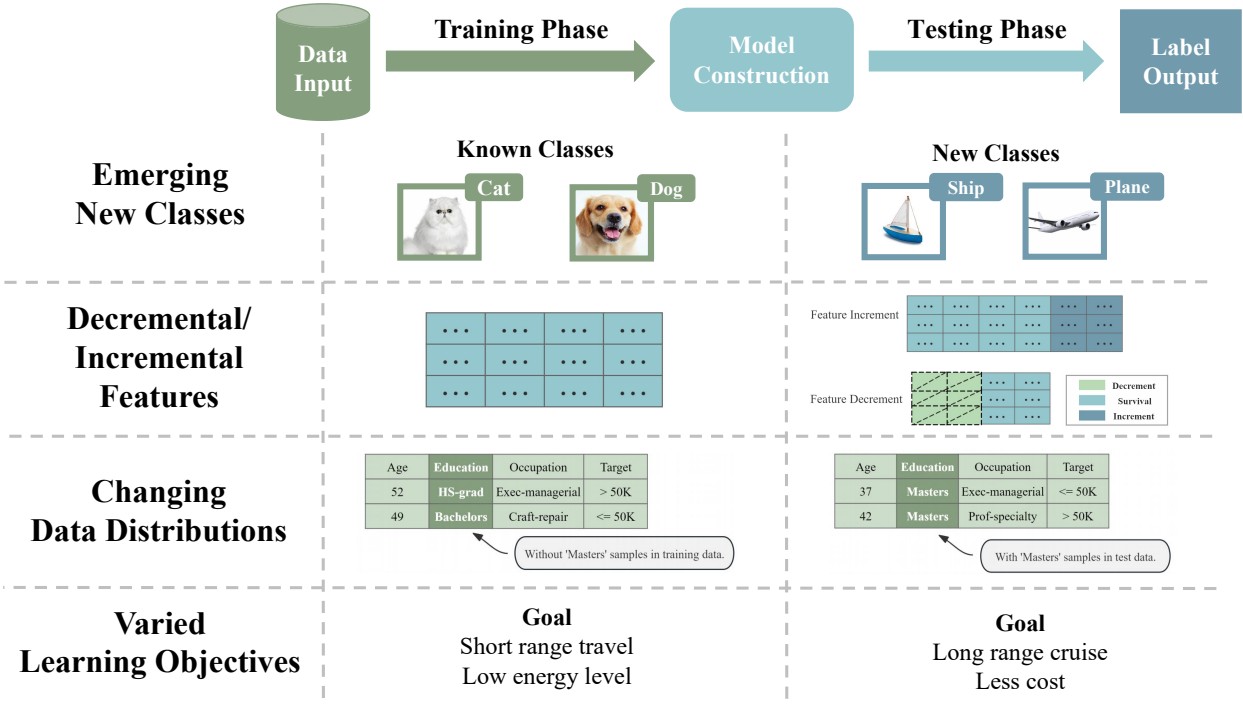

*Figure 1.* Open environments challenges include **emerging new classes, decremental/incremental features, changing data distributions**, and **varied learning objectives**.

environments, these studies mainly concentrate on distribution shift scenarios (Hoo et al., 2025b; Helli et al., 2024) and have not yet conducted a comprehensive evaluation of various challenges that TabPFN v2.5 may face in open environments. This limitation raises the natural question of **whether TabPFN v2.5 can maintain good performance in open environments**, and highlights the need for a more holistic assessment of TabPFN v2.5 in diverse and dynamic real-world scenarios.

To this end, we conduct a comprehensive evaluation of the performance of TabPFN v2.5 compared with various tabular machine learning models in open environments for the first time. Existing tabular benchmarks in open environments primarily evaluate models in isolated scenarios, limiting their methodological applicability to broader real-world tasks. To address this, we introduce a unified evaluation framework that systematically benchmarks diverse tabular models across challenges in open environments, enabling standardized assessment of robustness and adaptability.

From the experiments, we observe that **TabPFN v2.5 exhibits overall limitations across data-fitting challenges in open environments**. Although TabPFN v2.5 demonstrates strong capability in emerging new class detection and achieving consistent excellent performance across different learning objectives, which demonstrate its ability of prior-driven decision, it exhibits heightened vulnerability to decremental features, particularly in regression tasks, where

its performance gap widens considerably as feature shift increases, and it cannot handle newly added features at test time due to its reliance on fixed input dimensionality. Under changing data distributions, TabPFN v2.5 handles covariate shift relatively well but suffers substantial performance degradation under concept drift.

The above observations suggest that settings where TabPFN v2.5 is most likely the right choice in open environments for practitioners: 1) when the distribution shift is characterized as covariate shift; 2) when the label distribution is approximately imbalanced across classes; 3) when consistent performance across varied learning objectives is required.

Separately, despite their strong performance in closed environments, various models often exhibit limited generalization to open environments. This underscores a crucial research imperative regarding the enhancements required to advance open environments research. To address this, we propose the following recommendations:

- Develop more benchmarks and evaluation protocols targeting open environments.
- Evaluate models on various open environments metrics, especially robustness.
- Design specific modules like dynamic feature adaptation methods for TabPFN v2.5.
- Expose models to open environments via large-scale synthetic data during training.

## 2. Open Environments Challenges

In this section, we draw upon the previous work presented in (Zhou, 2022) as a foundational framework to further symbolically formalize and represent open environments challenges encountered in the tabular machine learning tasks.

### 2.1. Emerging New Classes

In a closed-environment setting, machine learning models assume that any test sample's class is restricted to the categories seen during training. However, this assumption does not always hold in open environments. We formally define this challenge by partitioning the class set $L$ into $L^{\text{train}}$ and $L^{\text{test}}$, corresponding to the training and testing phases, respectively. In closed environments, the class set remains consistent between the training and testing phases, i.e., $L^{\text{train}} = L^{\text{test}}$. In contrast, in open environments, test samples may belong to novel classes $l$ that are not present during training, i.e., $\exists\, l \in L^{\text{test}}$ such that $l \notin L^{\text{train}}$. In such cases, the model must be capable of identifying and handling these new classes.

### 2.2. Decremental/Incremental Features

Decremental and incremental features represent partial removal or augmentation of the input feature set, known as feature shift. Let $C$ denote the full feature set, partitioned into $C^{\text{train}}$ and $C^{\text{test}}$ for training and testing, respectively. In closed environments, $C^{\text{train}} = C^{\text{test}}$, whereas in open environments, $C^{\text{train}}$ remains fixed but $C^{\text{test}}$ may differ. Specifically, when $C^{\text{test}} \subsetneq C^{\text{train}}$, imputation of shifted features in $C^{\text{test}}$ is necessary to maintain input dimension consistency and enable accurate model prediction (**Decremental Features**). Conversely, when $C^{\text{train}} \subsetneq C^{\text{test}}$, models typically truncate the newly added features in $C^{\text{test}}$, retaining only those corresponding to $C^{\text{train}}$, thus ensuring input dimension consistency (**Incremental Features**).

### 2.3. Changing Data Distributions

Machine learning research in closed environments generally assumes that the data in both the training and testing phases are independent and identically distributed (i.i.d.). However, this assumption does not always hold in open environments. **Covariate Shift** (Sugiyama et al., 2007) occurs when the input distribution $p(x)$ changes between training and testing phases, while the conditional probability $p(y|x)$ remains constant. **Concept Shift** (Gama et al., 2014) involves changes in the conditional probability $p(y|x)$ with a stable input distribution $p(x)$.

### 2.4. Varied Learning Objectives

The performance of the machine learning model $f$ can be measured by a learning objective $M_f$, such as accuracy, F1 score, or ROC-AUC. Learning towards different objectives may lead to a model with different strengths. A model that is optimal on one measure is not necessarily optimal on others. Research in closed environments generally assumes that the $M_f$ used to evaluate model performance is fixed and known in advance. However, this assumption may not always hold in open environments. When facing this challenge, the model should perform well across various learning objectives without requiring data to be recollected and a completely new model to be trained.

## 3. Experiments

We undertake a comprehensive evaluation in open environments to rigorously assess the robustness and adaptability of TabPFN v2.5 through our proposed evaluation framework. Specifically, we subject TabPFN v2.5 to evaluation across four distinct challenges in open environments. We choose RandomForest (Breiman, 2001), XGBoost (Chen & Guestrin, 2016) and CatBoost (Prokhorenkova et al., 2018) as tree-based baseline models. We also select MLP, RealMLP (Holzmüller et al., 2025), and ModernNCA (Ye et al., 2024) as deep learning baseline models. Datasets selected for challenges adhere to the corresponding benchmark specifications to ensure consistency and fair comparison.

### 3.1. Main Results

Based on the evaluation framework we have proposed, which can be found in Appendix C, we assess various models on four challenges in open environments. We provide the Wilcoxon test in Figure 2. Experimental details and results are shown in Appendix F.

**Emerging New Classes: TabPFN v2.5 has the potential to detect new classes.** TabPFN v2.5 achieves the highest AUC across all datasets for new class detection, with significant improvements over other methods ($p < 0.001$). This stems from its pretraining on synthetic data, which enables better uncertainty estimation for emerging classes.

**Decremental/Incremental Features: TabPFN v2.5 exhibits heightened vulnerability to decremental features in regression tasks.** TabPFN v2.5's performance gap widens significantly with increasing feature shifts in regression tasks, indicating weaker adaptability and higher sensitivity. In contrast, MLP models exhibit greater robustness against decremental features, possibly due to their inherent anti-shift properties, which TabPFN v2.5 may lack.

**Changing Data Distributions: TabPFN v2.5 shows heterogeneous behavior under distribution shift.** TabPFN v2.5 handles covariate shift well but degrades under concept shift, indicating its priors help with $p(x)$ shifts but not $p(y \mid x)$. It shows no statistical advantage over competitors.

*Table 1.* Average rank across four open environments tasks. Ranks for emerging new classes are based on Accuracy over three uncertainty intervals. Ranks for decremental/incremental features are based on Accuracy and RMSE, while for changing data distributions and varied learning objectives are computed as the average rank across all four metrics. The best rank is in **bold** and the second rank is underlined.

| Challenge | RandomForest | XGBoost | CatBoost | MLP | RealMLP | ModernNCA | TabPFN v2.5 |
|---|---|---|---|---|---|---|---|
| Emerging New Classes | 2.00 | 5.92 | 4.58 | 5.17 | 4.92 | 4.08 | **1.25** |
| Decremental/Incremental Features | 5.27 | 4.07 | **2.00** | 4.53 | 4.20 | 5.07 | 2.87 |
| Changing Data Distributions | 3.89 | 3.89 | **2.00** | 6.89 | 4.44 | 4.11 | 2.56 |
| Varied Learning Objectives | 5.25 | 3.25 | 2.75 | 6.75 | 4.50 | 4.25 | **1.25** |
| Average Rank | 4.06 | 4.01 | 2.90 | 5.36 | 4.10 | 4.50 | **2.48** |

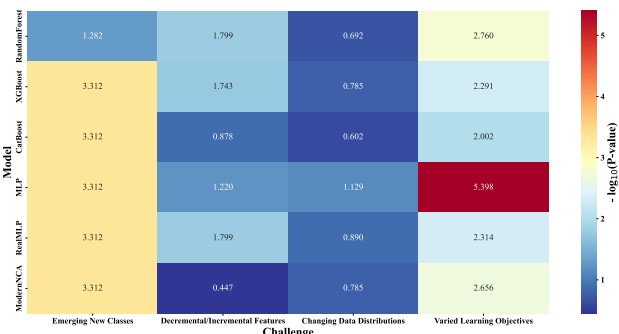

*Figure 2.* $Log_{10}$(P-value) of six models compared with TabPFN v2.5 across four challenge dimensions. Higher values indicate stronger evidence of superiority, with values exceeding 1.30 denoting statistical significance.

**Varied Learning Objectives: TabPFN v2.5 maintains consistently competitive performance across various learning objectives.** TabPFN v2.5 demonstrates state-of-the-art performance with respect to accuracy and ROC-AUC. More importantly, a comparative analysis reveals statistically significant performance advantages in F1 Score and Balanced Accuracy when contrasted with other models.

### 3.2. Holistic Assessment

Our comprehensive evaluation reveals that TabPFN v2.5's performance in open environments reflects a fundamental paradigm shift, showing a distinct dichotomy between semantic-level adaptations and data-level perturbations. **TabPFN v2.5 is better understood as a prior-driven decision model rather than a purely data-fitting tabular predictor.** It demonstrates strong adaptability to semantic-level changes like emerging new classes and varied learning objectives via prior-driven global inference, yet exhibits expected limitations including fixed input dimensionality, performance degradation under concept drift, and majority-class bias. Surprisingly, despite its majority bias, it achieves state-of-the-art performance on class-imbalance-sensitive metrics and new class detection, while showing unexpected vulnerability to decremental features, especially in regression tasks. **TabPFN v2.5 and tree-based models do not compete along the same dimension, while they represent two fundamentally distinct paradigms of intelligence.** As shown in Table 1, tree-based models excel in feature shifts

and unstable input schemas through feature-driven local partitioning, whereas TabPFN v2.5 excels in semantic-level adaptation via prior-driven global inference.

**Insights and implications.** Beyond rankings, TabPFN v2.5 cannot handle newly added features, degrades substantially under concept shift while resisting covariate shift, and shows strong new class detection but heightened vulnerability to decremental features in regression tasks; therefore, model selection must align with the specific capabilities prioritized by the target open environment scenario.

### 3.3. Recommendations

During the experimental investigation, we observed that existing high-performance models excel in closed environments but struggle with real-world open-environment challenges. To address this, we propose four recommendations: 1) **Develop comprehensive benchmarks and evaluation protocols for open environments tabular learning** by integrating diverse tasks and specialized metrics beyond traditional distribution shifts; 2) **Take model robustness as a critical metric when comparing model quality** alongside closed-environment performance; 3) **Design specific modular architecture and adaptive mechanisms for TabPFN v2.5** including dynamic feature adaptation, online adaptation modules for concept drift, multi-objective prediction heads, and scale-aware routing; 4) **Simulate open environments in the pre-training phase via large-scale synthetic data generation** using configurable change operators to expose models to open-environment conditions during training and establish standardized evaluation benchmarks.

## 4. Conclusion

We present the first comprehensive evaluation of TabPFN v2.5 in open environments and construct an evaluation framework that simulates diverse challenges in open environments, revealing its limitations in feature decrements and distribution shifts, while highlighting its strengths in detecting new classes and handling varied learning objectives. Empirical evaluations are constrained by restricted dataset diversity, compounded by insufficient theoretical depth, limiting both the fidelity of open environments simulation and the mechanistic understanding of TabPFN v2.5's robustness.

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

The Appendix consists of six sections:

- **Appendix A**: We review related work that complements the discussion in the main text.

- **Appendix B**: We provide the preliminary and additional details about TabPFN.

- **Appendix C**: We introduce the unified evaluation framework for open environments challenges.

- **Appendix D**: We provide detailed descriptions of the datasets used in our experiments, along with the differences from our framework.

- **Appendix E**: We provide the experimental setups and evaluated models.

- **Appendix F**: We include complete experimental results that were omitted from the main paper due to space limitations.

## A. Related Work

### A.1. Open Environments Challenges

Most tabular machine learning models are typically trained and tested in closed environments where critical learning factors remain stable. However, various real-world tasks operate in open environments where dynamic changes occur in key factors, posing challenges to model generalization (Parmar et al., 2023). Zhou (2022) categorizes four core challenges in open environments: Emerging New Classes, Decremental/Incremental Features, Changing Data Distributions, and Varied Learning Objectives.

These challenges are pivotal for machine learning in open environments. Emerging new classes, involving unseen classes during testing, have been addressed in natural language processing (Deng et al., 2024) and computer vision (Dhamija et al., 2020; Du et al., 2022). Decremental/Incremental features, caused by changes in feature sets, lead to mismatched training-testing spaces. TabFSBench (Cheng et al., 2025) evaluates model performance under such variations, and Hou et al. (2017) enhances performance by restoring ephemeral features. Changing data distributions, where test data violate the i.i.d. assumption, have led to benchmark datasets such as Tableshift (Gardner et al., 2023), and methods such as domain adaptation (Zhou et al., 2025) and domain generalization (Zhou et al., 2021). Varied learning objectives, which prioritize adaptive optimization beyond accuracy, include multi-objective learning (Zhou et al., 2019; Zuluaga et al., 2013) and self-evolving training (Liu et al., 2024b). However, research on these challenges remains fragmented, lacking a unified framework to evaluate models on all four challenges.

### A.2. Tabular Data in Machine Learning

Tabular data, with structured and heterogeneous features, is used in healthcare, finance, and recommendation systems (Borisov et al., 2022; Kadra et al., 2021). Unlike images and texts, it has high dimensionality, and complex dependencies, posing challenges for machine learning models (Fang et al., 2024). Current approaches are mainly tree-based models (e.g., XGBoost (Chizat et al., 2020), LightGBM (Badirli et al., 2020),CatBoost (Prokhorenkova et al., 2018)) and deep learning models. Tree-based models handle irregular patterns and uninformative features well (Grinsztajn et al., 2022), while deep learning models like FT-Transformer (Gorishniy et al., 2021) and NODE (Popov et al., 2020) aim to capture complex feature interactions for better performance (Gorishniy et al., 2021; Popov et al., 2020).

In the realm of tabular machine learning tasks, tree-based models have traditionally held a dominant position over deep learning models (Grinsztajn et al., 2022; McElfresh et al., 2023). The emergence of the novel deep learning model TabPFN v2 (Hollmann et al., 2025) has surpassed tree-based models. TabPFN v2 has demonstrated superior performance compared to tree-based models across multiple benchmarks. However, the majority of these benchmarks are confined to closed environments. Consequently, the comparative performance of TabPFN v2 and tree-based models in open environments remains underexplored.

### A.3. Research on TabPFN

TabPFN (Hollmann et al., 2023), short for Tabular Prior-Fitted Network, is a model pre-trained on large-scale synthetic datasets, enabling efficient zero-shot learning. It can efficiently perform classification and regression tasks without the need for hyperparameter tuning. Compared to existing models, TabPFN shows significant advantages on small- to medium-scale

datasets with low computational cost, making it an efficient solution for tabular tasks. Recent research, however, reveals limitations in TabPFN's performance on high-dimensional, large-scale, or multi-class tasks (Ye et al., 2025; Liu & Ye, 2025).

Various optimization strategies have been proposed to enhance TabPFN's adaptability to current limitations and more complex scenarios, including local context construction via retrieval-based methods (Koshil et al., 2024), model fine-tuning (Thomas et al., 2024; Xu et al., 2025), and pretraining dataset expansion (Breejen et al., 2024). Moreover, TabPFN's strong performance has prompted its application to challenges such as distribution shift adaptation (Helli et al., 2024), time series forecasting (Hoo et al., 2025a), and various domains including healthcare (Noda et al., 2024; Tran & Byeon, 2024), ecology (Heinzel et al., 2025), and cybersecurity (Ruiz-Villafranca et al., 2024). However, these studies primarily focus on closed environments or address only a single challenge in open settings, without providing a comprehensive evaluation of TabPFN under diverse open-environment scenarios.

## B. TabPFN and Its Variants

This section explains how TabPFN v1, TabPFN v2, and TabPFN v2.5, work. Since there is already a variety of research on these models, we provide a brief summary here.

### B.1. TabPFN v1

TabPFN v1 (Hollmann et al., 2023) formulates tabular classification as a sequence modeling task solvable via in-context learning (ICL) using a pre-trained Transformer (Vaswani et al., 2017). TabPFN encodes each training sample $(x_i, y_i)$ as a vector $(\tilde{x}_i, \tilde{y}_i)$ via linear projection and zero-padding, yielding uniform dimensionality in a $k$-dimensional space. A context matrix $\mathcal{A}$ is formed by concatenating $N$ such training samples along with a query test sample $x^*$:

$$\mathcal{A} = [\tilde{x}_i \oplus \tilde{y}_i]_{i=1}^N \parallel [\tilde{x}^*],$$

where $\oplus$ denotes vector concatenation. This representation treats all samples as tokens in a sequence, enabling the Transformer to infer the label of the query sample by attending to the context. The final class probabilities are generated by an MLP head applied to the output embedding of $x^*$. TabPFN is trained exclusively on synthetic tasks sampled from structural causal models, allowing it to approximate Bayesian inference across diverse tabular distributions without relying on real data during training.

### B.2. TabPFN v2

TabPFN v2 (Hollmann et al., 2025) extends the original architecture with innovations enhancing heterogeneous feature representation and complex tabular domain generalization. While TabPFN introduced ICL and synthetic task training for tabular data, TabPFN v2 emphasizes architectural generality, feature robustness, and scalability. A core innovation is randomized feature tokens, where each input feature is projected into a $k$-dimensional latent space with controlled noise, eliminating dedicated embeddings while preserving feature identity (Ye et al., 2025; Gorishniy et al., 2021). This design enables 3D tensor processing (samples × features × dimensions) with dual attention mechanisms, including cross-sample attention for capturing inter-example patterns and intra-feature attention for modeling column dependencies.

### B.3. TabPFN v2.5

TabPFN v2.5 (Grinsztajn et al., 2025) is the newest version of TabPFN, delivering major advances in scalability, performance, and deployment flexibility. It supports datasets with up to 50,000 rows and 2,000 features, representing a 20× increase in data capacity compared to TabPFN v2. The model architecture has been deepened to 18–24 layers, introduces learned "thinking" rows, and achieves 1× to 2.3× faster inference. On the TabArena benchmark, the default configuration outperforms tuned tree-based models and approaches the accuracy of AutoGluon 1.4, a complex ensemble tuned for four hours.

Current research (Grinsztajn et al., 2025; Ye et al., 2025) has extensively evaluated TabPFN's performance in closed environments, but largely overlooked its adaptability to open environments, leaving a critical gap. To fully realize its practical value, we conduct comprehensive evaluations of TabPFN v2.5 under various open environments challenges.

## C. Evaluation Framework for Open Environments Challenges

Existing benchmarks for evaluating model performance in open environments typically focus on a single task, such as distribution shift (Gardner et al., 2023) or feature shift (Cheng et al., 2025). They lack a unified and comprehensive assessment across multiple open environments challenges. Hence, we propose a modular and extensible evaluation framework that assesses both model performance and robustness across diverse real-world scenarios. The framework formalizes four representative open-environment challenges, namely Emerging New Classes, Decremental and Incremental Features, Changing Data Distributions, and Varied Learning Objectives. It builds testing protocols by leveraging existing benchmarks, including WhyShift (Liu et al., 2023) and TableShift (Gardner et al., 2023) for distribution shifts, and TabFSBench (Cheng et al., 2025) for feature shifts. It also supports comprehensive evaluation of tabular models and enables exporting datasets under different open environments scenarios within a few lines of Python code.

To facilitate the use of our proposed evaluation framework, we provide a set of APIs. More details are available at **https://anonymous.4open.science/r/tabpfn-ood-4E65**. The API accepts four parameters: `dataset`, `model`, `task`, and `export_dataset`. We will give further specifications in the repository README.md.

The `dataset` parameter specifies the full name of the dataset to be used. Our evaluation framework supports datasets from OpenML, Kaggle, and local directories.

The `model` parameter defines the model to be evaluated and can be selected from a range of tree-based models and deep-learning models, which are systematically evaluated in this paper. Additional new models can also be flexibly incorporated by following the detailed instructions provided in the "How to Add New Models" section.

The `task` parameter determines the type of experiments in open environments to be conducted. The available options include emerging new classes (enc), decremental features (df), changing data distributions (cdd), and varied learning objectives (vlo).

The `export_dataset` parameter controls whether the modified dataset, corresponding to a specific open environments challenge, is exported as a CSV file for further use or analysis.

An example command for running the evaluation framework is as follows:

**Example Command**

```
python run.py --dataset Adult --model xgboost --task enc --export_dataset
True
```

## D. Existing Benchmarks and Distinctions from Our Framework

To contextualize our evaluation framework, we review three widely used benchmarks (TabFSBench, WhyShift, and TableShift) and highlight key differences from our unified evaluation framework.

### D.1. Overview of Existing Benchmarks

**TabFSBench** TabFSBench (Cheng et al., 2025) is a benchmark specifically designed to evaluate model robustness under feature shift, where features are partially removed or perturbed at test time. It includes four distinct feature shift settings across twelve datasets (eight classification and four regression tasks). TabFSBench systematically analyzes how varying degrees of feature loss affect model performance.

**WhyShift** WhyShift (Liu et al., 2023) focuses on evaluating model robustness under distributional shifts, with an emphasis on $Y|X$-shifts. It includes five tabular datasets spanning 22 source-target domain pairs and employs the DISDE framework for fine-grained, shift-specific analysis. The benchmark highlights high-risk covariate regions and supports both algorithmic and data-level interventions. Although organized under a common framework, WhyShift applies distinct evaluation protocols tailored to each shift type.

**TableShift** TableShift (Gardner et al., 2023) provides a collection of 15 binary tabular classification tasks with predefined source-target domain pairs for evaluating classical domain adaptation and generalization. The benchmark spans real-

world domains including finance, education, public policy, healthcare, and civic engagement. It standardizes data access, preprocessing, partitioning, and evaluation via a unified Python API, and supports systematic analysis of various types of distributional shift, such as label and covariate shift. All tasks are formulated in a closed-set setting, where test classes are identical to those observed during training.

### D.2. Our Unified Evaluation Framework

While our framework builds upon insights from existing benchmarks, it introduces a higher degree of integration, extensibility, and analytical capability:

- Joint evaluation of four open environments challenges: In contrast to prior benchmarks that target individual shift types, our framework jointly supports feature shift, domain shift, class shift, and task adaptation within a unified and configurable platform.

- Standardized evaluation protocol: We provide a consistent pipeline for data preprocessing, model interfacing, and metric reporting, enabling reproducible and comparable evaluations across heterogeneous distribution shifts.

- Modular and extensible design: The framework supports plug-and-play integration of new datasets, models, and adaptation strategies, facilitating efficient experimentation and ablation studies.

- Holistic generalization analysis: By assessing multiple shift types within a single system, the framework enables systematic analysis of model generalization capabilities that are not jointly supported by existing benchmarks.

In summary, while our framework integrates components from existing benchmarks, it distinguishes itself by offering a unified, extensible, and task-complete evaluation environment. This design enables coherent and efficient analysis of model robustness across diverse open-environment scenarios, offering methodological capabilities that surpass the sum of its component benchmarks.

## E. General Experimental Settings

### E.1. Traning settings

Deep learning models are trained on an NVIDIA 4090 GPU. Tree-based models are trained on an AMD Ryzen 5 7500F 6-Core Processor. All results are reported as the average of three different random seeds to ensure statistical reliability.

### E.2. Models

In this subsection, we provide detailed descriptions of all the models used in our paper.

**XGBoost**   XGBoost (Chizat et al., 2020) is an efficient and flexible machine learning model that incrementally builds multiple decision trees by optimizing the loss function, with each tree correcting the errors of the previous one to continuously improve the model's predictive performance. XGBoost also incorporates the gradient boosting algorithm, iteratively training decision tree-based models with the goal of minimizing residuals and enhancing predictive accuracy.

**CatBoost**   CatBoost (Prokhorenkova et al., 2018) is a powerful boosting-based model designed for efficient handling of categorical features. It uses the "Ordered Boosting" technique, which calculates gradients sequentially to prevent target leakage and maintain the independence of each training instance. At the same time, CatBoost employs "Target-based Categorical Encoding," converting categorical variables into numerical representations based on target statistics, thereby reducing the need for extensive preprocessing and improving model performance.

**RandomForest**   RandomForest (Breiman, 2001) is a classical ensemble learning method based on bagging and decision trees. It constructs a multitude of decision trees during training and outputs the mode or mean prediction of individual trees. Its robustness to overfitting, strong performance with minimal tuning, and ability to handle both classification and regression tasks make it a widely used baseline in tabular data benchmarks.

*Table 2.* Hyperparameter Grids of Tree-based Models.

| Model | Hyperparameter | Values |
|---|---|---|
| **XGBoost** | Learning Rate | $\{0.01, 0.1\}$ |
| | Max. Depth | $\{1, 5, 9\}$ |
| | N Estimators | $\{10000, 20000, 30000\}$ |
| | Subsample | $\{0.5, 0.8, 1.0\}$ |
| | Colsample Bytree | $\{0.5, 0.8, 1.0\}$ |
| | Min Child Weight | $\{1, 3, 5\}$ |
| **CatBoost** | Learning Rate | $\{0.01, 0.05, 0.1\}$ |
| | Depth | $\{4, 6, 8\}$ |
| | Iterations | $\{500, 1000, 2000\}$ |
| **RandomForest** | Min Samples Split | $[2, 10]$ |
| | Min Samples Leaf | $[1, 10]$ |

**MLP**   An MLP consists of multiple layers of neurons, with each layer fully connected to the next. An MLP contains at least three layers: an input layer, one or more hidden layers, and an output layer. It continuously adjusts the connection weights between neurons through training methods such as the backpropagation algorithm and gradient descent to minimize prediction errors.

**ModernNCA**   ModernNCA (Ye et al., 2024) is an enhanced Neighborhood Component Analysis (NCA) model that improves tabular data processing by adjusting learning objectives, integrating deep learning models, and using stochastic neighbor sampling for better efficiency and accuracy.

**RealMLP**   RealMLP (Holzmüller et al., 2024) is an enhanced multilayer perceptron designed for tabular data tasks, combining architectural improvements with meta-learned default hyperparameters. It achieves a strong balance between accuracy and training efficiency.

### E.3. Hyperparameter Tuning

In this subsection, we provide hyperparameter grids of tree-based and deep learning models in Table 2, 3.

For tree-based models, we employ GridSearchCV from the scikit-learn library to conduct an exhaustive hyperparameter search. This approach systematically explores a predefined parameter grid through 5-fold cross-validation to ensure the reproducibility of results. The search process is optimized for computational efficiency by enabling parallel processing.

Regarding deep learning models, we implement an adaptive hyperparameter optimization strategy based on the Optuna framework (Akiba et al., 2019), following methodologies established in prior studies (Liu et al., 2024a). The optimization protocol maintains a constant batch size of 1024 and performs 100 independent trials using training-validation splits to prevent potential data leakage from the test set.

## F. Comprehensive Evaluation of TabPFN v2.5

We undertake a comprehensive evaluation in open environments to rigorously assess the robustness and adaptability of TabPFN v2.5 through our proposed evaluation framework. Specifically, we subject TabPFN v2.5 to evaluation across four distinct challenges in open environments. We choose RandomForest (Breiman, 2001), XGBoost (Chen & Guestrin, 2016) and CatBoost (Prokhorenkova et al., 2018) as tree-based baseline models. We also select MLP, RealMLP (Holzmüller et al., 2025), and ModernNCA (Ye et al., 2024) as deep learning baseline models. Datasets selected for different challenges adhere to the corresponding benchmark specifications to ensure consistency and fair comparison.

### F.1. Emerging New Classes

In the tabular domain, benchmark protocols for evaluating the novel class detection capability of tabular models remain largely unexplored. To fill this gap, we construct a novel class detection task for tabular data following an MSP-based evaluation paradigm inspired by prior out-of-distribution detection studies (Hendrycks & Gimpel, 2017). Specifically, for a

*Table 3.* Hyperparameter Grids of Deep Learning Models.

| Model | Hyperparameter | Values |
|---|---|---|
| **MLP** | D_layers | $\{1, 8, 64, 512\}$ |
| | Dropout | Uniform $\{0.0, 0.5\}$ |
| | Learning Rate | Loguniform$\{e^{-5}, 0.01\}$ |
| | Weight Decay | Loguniform$\{e^{-6}, 0.001\}$ |
| **ModernNCA** | Dropout | Uniform $\{0.0, 0.5\}$ |
| | D_block | Int$\{64, 1024\}$ |
| | N_blocks | Int$\{0, 2\}$ |
| | N_frequencies | Int$\{16, 96\}$ |
| | Frequency Scale | Loguniform$\{0.005, 10\}$ |
| | D_embedding | Int$\{16, 64\}$ |
| | Sample Rate | Uniform$\{0.05, 0.6\}$ |
| | Learning Rate | Loguniform$\{e^{-5}, 0.1\}$ |
| | Weight Decay | Loguniform$\{e^{-6}, 0.001\}$ |
| **RealMLP** | Num Emb Type | \{none, pbld, pl, plr\} |
| | Add Front Scale | \{True, False\} |
| | Learning Rate (lr) | $\log U(0.02, 0.3)$ |
| | Dropout (p_drop) | \{0.00, 0.15, 0.30\} |
| | Activation (act) | \{selu, relu, mish\} |
| | Hidden Sizes | $\{[256, 256, 256], [64, 64, 64, 64, 64], [512]\}$ |
| | Weight Decay (wd) | \{0.0, 0.02\} |
| | PLR Sigma | $\log U(0.05, 0.5)$ |
| | Label Smoothing Epsilon (ls_eps) | \{0.0, 0.1\} |

$k$-class dataset, we employ a leave-one-class-out protocol and conduct $k$ runs, each time excluding one class from training and treating its samples as novel at test time, while the remaining classes are regarded as known. After training on the reduced dataset, we compute the Maximum Softmax Probability (MSP) for each test sample as a confidence score. We define a predefined uncertainty interval $[\theta_{\min}, \theta_{\max}]$ and classify samples whose MSP falls within this interval as novel, as such predictions indicate uncertainty with respect to the known classes. Since this task is essentially a binary detection problem and often involves class imbalance, we report Accuracy, ROC-AUC, and AUPR, with results averaged over all runs.

**TabPFN v2.5 has the potential to detect new classes.** As illustrated in Table 4, TabPFN v2.5 consistently achieves better AUC in the task of new class detection across all four datasets when compared to other models. The Wilcoxon signed-rank test further confirms that these improvements are statistically significant ($p < 0.001$) compared to most methods, except Random Forest. This advantage can be attributed to the pretraining paradigm of TabPFN v2.5. Pretrained on massive synthetic tabular data and diverse classification tasks generated from rich structural priors, the model learns a transferable global prior, which allows it to produce more discriminative predictive uncertainty for samples from emerging classes. This property is crucial for new class detection.

### F.2. Decremental/Incremental Features

To conduct a comprehensive evaluation of decremental/incremental features, we adopt twelve datasets in TabFSBench across various domains and feature types. Specifically, we randomly remove 20%, 40%, 60%, 80%, and 100% of the features from $C^{\text{test}}$, constructing varying levels of feature shifts.

**TabPFN v2.5 exhibits heightened vulnerability to decremental features in regression tasks.** Tables 5 shows that TabPFN v2.5's performance gap widens significantly with increasing feature shifts in regression tasks, indicating weaker adaptability and higher sensitivity. In contrast, MLP models exhibit greater robustness against decremental features, possibly due to their inherent anti-shift properties, which TabPFN v2.5 may lack. The Wilcoxon test results presented in Figure 5 also provide further statistical support for this conclusion.

**TabPFN v2.5 cannot handle newly added features at test time.** When input dimensionality increases dynamically, TabPFN v2.5 is unable to process new features and instead truncates them, keeping only those observed during training. This is due to its reliance on fixed feature dimensionality in both parameterization and internal representation. Nevertheless, this limitation does not negatively impact its performance.

*Table 4.* Average ACC, AUC, and AUPR for emerging class detection based on prediction uncertainty, computed over three uncertainty intervals ([0.4,0.6], [0.45,0.55], and [0.49,0.51]). Higher is better; the best result is **bold** and the second best is underlined.

| Metrics | Dataset | RandomForest | XGBoost | CatBoost | MLP | RealMLP | ModernNCA | TabPFNv2.5 |
|---|---|---|---|---|---|---|---|---|
| ACC | EyeMovement | 0.394 | 0.058 | 0.081 | 0.015 | 0.103 | 0.054 | **0.510** |
| | CMC | 0.308 | 0.083 | 0.099 | 0.199 | 0.283 | 0.180 | **0.516** |
| | Wine-R | 0.348 | 0.030 | 0.056 | 0.041 | 0.000 | 0.095 | **0.527** |
| | Wine-W | 0.287 | 0.025 | 0.037 | 0.041 | 0.000 | 0.228 | **0.488** |
| AUC | EyeMovement | 0.509 | 0.503 | 0.508 | 0.503 | 0.510 | 0.504 | **0.510** |
| | CMC | 0.502 | 0.502 | 0.512 | **0.527** | 0.514 | 0.510 | 0.516 |
| | Wine-R | 0.500 | 0.567 | 0.467 | **0.700** | 0.500 | 0.400 | 0.527 |
| | Wine-W | 0.500 | **0.533** | 0.367 | 0.400 | 0.500 | 0.500 | 0.488 |
| AUPR | EyeMovement | 0.505 | 0.501 | 0.504 | 0.502 | **0.506** | 0.502 | **0.506** |
| | CMC | 0.502 | 0.501 | 0.507 | **0.517** | 0.508 | 0.505 | 0.510 |
| | Wine-R | 0.500 | 0.558 | 0.492 | **0.658** | 0.500 | 0.481 | 0.520 |
| | Wine-W | 0.500 | **0.533** | 0.462 | 0.472 | 0.500 | 0.500 | 0.499 |

*Table 5.* Average performance and performance gap across different tasks under decremental features. We simulate feature shift by progressively altering the input feature set at five levels: 20%, 40%, 60%, 80%, and 100%. For each level of shift, we compute the model's performance gap relative to the original (0%) setting, separately for each task. For classification tasks, we report accuracy (higher is better), and for regression tasks, RMSE (lower is better). Each cell reports $x\ (\pm y)$, where $x$ denotes the performance under the given shift level, and $y$ represents the gap compared to the original performance. The best performance is shown in **bold**, and the smallest gap is underlined.

| Task | Shift | RandomForest | XGBoost | CatBoost | MLP | RealMLP | ModernNCA | TabPFN v2.5 |
|---|---|---|---|---|---|---|---|---|
| Binary Classification | 0% | 0.838 | 0.842 | **0.869** | 0.805 | 0.813 | **0.869** | 0.852 |
| | 20% | 0.764(-0.074) | 0.766(-0.076) | **0.834(-0.035)** | 0.781(-0.024) | 0.744(-0.069) | 0.708(-0.161) | 0.815(-0.037) |
| | 40% | 0.622(-0.216) | 0.624(-0.218) | 0.764(-0.105) | 0.743(-0.062) | 0.666(-0.147) | 0.598(-0.271) | **0.768(-0.084)** |
| | 60% | 0.583(-0.255) | 0.581(-0.261) | 0.714(-0.155) | 0.698(-0.107) | 0.672(-0.141) | 0.568(-0.301) | **0.732(-0.120)** |
| | 80% | 0.464(-0.374) | 0.514(-0.328) | 0.631(-0.238) | 0.620(-0.185) | 0.563(-0.250) | 0.540(-0.329) | **0.709(-0.143)** |
| | 100% | 0.446(-0.392) | 0.467(-0.375) | 0.537(-0.332) | 0.534(-0.271) | 0.460(-0.353) | 0.512(-0.357) | **0.648(-0.204)** |
| Multi Classification | 0% | 0.800 | 0.802 | 0.837 | 0.723 | 0.745 | **0.906** | 0.801 |
| | 20% | 0.735(-0.065) | 0.759(-0.043) | 0.794(-0.043) | 0.700(-0.023) | 0.640(-0.105) | **0.819(-0.087)** | 0.801(-0.024) |
| | 40% | 0.637(-0.163) | 0.677(-0.125) | 0.714(-0.123) | 0.658(-0.065) | 0.665(-0.080) | 0.700(-0.206) | **0.741(-0.084)** |
| | 60% | 0.462(-0.338) | 0.574(-0.228) | 0.605(-0.232) | 0.600(-0.123) | 0.559(-0.186) | 0.562(-0.344) | **0.698(-0.127)** |
| | 80% | 0.354(-0.446) | 0.460(-0.342) | 0.463(-0.374) | 0.520(-0.203) | 0.379(-0.366) | 0.444(-0.462) | 0.413(-0.412) |
| | 100% | 0.226(-0.574) | 0.306(-0.496) | 0.321(-0.516) | 0.363(-0.360) | 0.195(-0.550) | 0.286(-0.620) | **0.419(-0.406)** |
| Regression | 0% | 0.925 | 0.922 | 0.902 | 0.997 | 0.926 | 0.940 | **0.803** |
| | 20% | 1.218(+0.293) | 1.155(+0.233) | 1.152(+0.250) | 1.025(+0.028) | 1.263(+0.337) | 1.103(+0.163) | **1.068(+0.265)** |
| | 40% | 1.537(+0.612) | 1.514(+0.592) | 1.544(+0.642) | 1.073(+0.076) | 1.567(+0.641) | 1.309(+0.369) | 1.349(+0.546) |
| | 60% | 1.738(+0.813) | 1.762(+0.840) | 1.818(+0.916) | **1.125(+0.128)** | 1.802(+0.876) | 1.499(+0.559) | 1.621(+0.818) |
| | 80% | 2.086(+1.161) | 2.119(+1.197) | 2.247(+1.345) | **1.181(+0.184)** | 2.138(+1.212) | 1.735(+0.795) | 1.752(+0.949) |
| | 100% | 2.346(+1.421) | 2.412(+1.490) | 2.571(+1.669) | **1.247(+0.250)** | 2.433(+1.507) | 1.940(+1.000) | 1.723(+0.920) |

## F.3. Changing Data Distributions

We evaluate TabPFN v2.5 under scenarios of changing data distributions, using metrics including Accuracy, Balanced Accuracy, F1-score, and ROC-AUC. The evaluation is conducted on nine fully numerical datasets drawn from the WhyShift (Liu et al., 2023) and TableShift (Gardner et al., 2023) benchmarks. Following the shift taxonomy provided by WhyShift, we further categorize the six datasets selected from WhyShift into two groups, covariate shift and concept shift, for analysis.

**TabPFN v2.5 shows heterogeneous behavior under distribution shift.** As shown in Table 6 and Figure 3, TabPFN v2.5 appears to handle covariate shift better than concept shift, achieving strong performance on several covariate shift benchmarks while exhibiting clear performance degradation under concept shift. This contrast may suggest that the global priors acquired during pretraining contribute to the robustness of TabPFN v2.5 under shifts in the input distribution $(p(x))$, while offering limited support for adaptation to shifts in the conditional distribution $(p(y \mid x))$. Moreover, the Wilcoxon test(See Figure 5) shows that TabPFN v2.5 has no statistically significant advantage over competing methods under distribution shift, suggesting that current pretraining paradigms have yet to learn sufficiently stable and transferable representations that generalize robustly under distribution shifts.

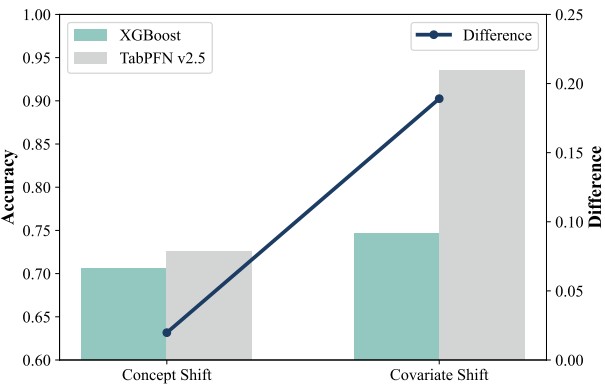

*Figure 3.* Model performance comparison on accuracy between TabPFN v2.5 and XGBoost on changing data distributions task.

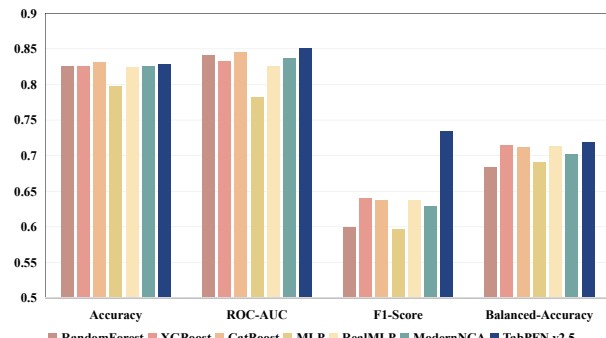

*Figure 4.* Model performance on four learning objectives of the classification task.

## F.4. Varied Learning Objectives

We conduct an exhaustive comparative analysis across four primary classification learning objectives, including Accuracy, ROC-AUC, F1-score, and Balanced Accuracy. The analysis is performed on i.i.d. datasets used in changing data distribution.

**TabPFN v2.5 demonstrates superior robustness to class imbalance.** Figure 4 reveals that TabPFN v2.5 achieves the highest performance on class-imbalance-sensitive metrics (F1-score and Balanced Accuracy), suggesting exceptional capability in handling minority classes. Specifically, Balanced Accuracy, a metric designed to address class imbalance by computing the arithmetic mean of per-class accuracy, shows that TabPFN v2.5 effectively adapts to varying sample sizes across different classes, outperforming all competing models by a noticeable margin. Similarly, F1-score, as the harmonic mean of precision and recall, further confirms the model's superior predictive capability for minority classes, with an improvement of approximately 30% over the second best.

**TabPFN v2.5 maintains consistently competitive performance across various learning objectives.** As illustrated in Figure 4, TabPFN v2.5 demonstrates state-of-the-art performance with respect to accuracy and ROC-AUC, achieving the highest ROC-AUC and comparable accuracy to top-performing models. More importantly, a comparative analysis reveals statistically significant performance advantages in F1 Score and Balanced Accuracy when contrasted with other models. These results highlight a key strength of TabPFN v2.5. Unlike conventional models that excel in specific evaluation criteria while faltering in others, TabPFN v2.5 maintains consistent efficacy across all evaluated performance metrics, establishing it as a robust general-purpose solution for tabular classification tasks. In addition, the Wilcoxon test results shown in Figure 5 further indicate that TabPFN v2.5 has a statistically significant advantage over other competing models under the Varied Learning Objectives setting.

## F.5. Holistic Assessment

Our comprehensive evaluation reveals that the performance of TabPFN v2.5 in open environments is not merely a matter of metric fluctuation, but a reflection of a fundamental shift in machine learning paradigms. We observe a distinct performance dichotomy between semantic-level adaptations and data-level perturbations.

**TabPFN v2.5 is better understood as a prior-driven decision model rather than a purely data-fitting tabular predictor.** This is evidenced by its strong adaptability to semantic-level changes, such as emerging new classes and varying learning objectives, with prior-driven global inference enabling high-level decision transfer. Consistent with its architectural constraints, TabPFN v2.5 exhibits expected limitations, including fixed input dimensionality that precludes handling of unseen features, performance degradation under concept drift, and a persistent majority-class bias. Yet several findings defy these expectations. Despite its majority bias, TabPFN v2.5 achieves state-of-the-art performance on class-imbalance-sensitive metrics (F1-score and balanced accuracy) and demonstrates surprising strength in new class detection, yet shows unexpected vulnerability to decremental features, particularly in regression tasks. This dichotomy reveals that its semantic-level strengths stem from prior-driven global inference, while its fragility under data-level perturbations reflects the inherent constraints of its fixed architecture, explaining why tree-based models maintain an advantage in unstable input scenarios through feature-driven local partitioning.

*Table 6.* Performance of evaluated models under distribution shift. The best results are highlighted in **bold**, and the second best are underlined.

| Metrics | Model | ACS Income (CA-PR) | ACS Mobility (MS-HI) | ACS Pub.Cov (NE-LA) | ACS Pub.Cov (2010-2017) | ACS Income (Setting 21) | ACS Income (Setting 22) | college scorecard | brfss diabetes | diabetes readmission |
|---|---|---|---|---|---|---|---|---|---|---|
| Accuracy | RandomForest | 0.697 | 0.729 | **0.715** | **0.706** | 0.729 | 0.739 | 0.846 | 0.828 | 0.608 |
| | XGBoost | 0.737 | 0.743 | 0.710 | 0.672 | 0.741 | 0.753 | 0.850 | 0.824 | 0.611 |
| | CatBoost | 0.675 | 0.724 | 0.694 | 0.692 | 0.729 | 0.745 | 0.860 | 0.829 | **0.622** |
| | MLP | 0.637 | 0.681 | 0.658 | 0.679 | 0.697 | 0.712 | 0.815 | 0.827 | 0.554 |
| | RealMLP | 0.653 | 0.675 | 0.678 | 0.679 | 0.707 | 0.721 | 0.838 | 0.828 | 0.616 |
| | ModernNCA | 0.655 | 0.692 | 0.689 | 0.687 | 0.719 | 0.733 | 0.848 | 0.829 | 0.611 |
| | TabPFN v2.5 | **0.895** | **0.786** | 0.594 | 0.629 | **0.928** | **0.944** | **0.862** | **0.834** | 0.621 |
| ROC-AUC | RandomForest | 0.829 | **0.723** | **0.738** | **0.733** | 0.768 | 0.787 | 0.872 | 0.815 | 0.667 |
| | XGBoost | 0.823 | 0.667 | 0.672 | 0.681 | 0.732 | 0.762 | 0.903 | 0.796 | 0.659 |
| | CatBoost | **0.833** | 0.712 | 0.734 | 0.731 | 0.779 | 0.801 | 0.911 | 0.812 | 0.680 |
| | MLP | 0.782 | 0.657 | 0.674 | 0.673 | 0.724 | 0.746 | 0.859 | 0.807 | 0.574 |
| | RealMLP | 0.821 | 0.695 | 0.706 | 0.709 | 0.747 | 0.768 | 0.875 | 0.809 | 0.665 |
| | ModernNCA | 0.821 | 0.722 | 0.736 | 0.731 | 0.769 | 0.789 | 0.902 | 0.810 | 0.666 |
| | TabPFN v2.5 | 0.809 | 0.498 | 0.665 | 0.593 | **0.947** | **0.940** | **0.922** | **0.817** | **0.682** |
| F1-Score | RandomForest | 0.353 | 0.608 | 0.548 | **0.481** | 0.532 | 0.564 | 0.696 | 0.076 | 0.487 |
| | XGBoost | 0.376 | **0.609** | 0.559 | 0.389 | 0.613 | 0.643 | 0.723 | 0.236 | 0.562 |
| | CatBoost | 0.349 | 0.580 | **0.568** | 0.368 | 0.620 | 0.649 | 0.735 | 0.224 | 0.558 |
| | MLP | 0.324 | 0.548 | 0.527 | 0.391 | 0.583 | 0.610 | 0.670 | 0.190 | 0.538 |
| | RealMLP | 0.331 | 0.571 | 0.531 | 0.473 | 0.530 | 0.566 | 0.713 | 0.180 | 0.563 |
| | ModernNCA | 0.333 | 0.585 | 0.534 | 0.473 | 0.536 | 0.574 | 0.722 | 0.160 | 0.528 |
| | TabPFN v2.5 | **0.604** | 0.440 | 0.373 | 0.387 | **0.824** | **0.796** | **0.826** | **0.577** | **0.611** |
| Balanced-Accuracy | RandomForest | 0.740 | 0.621 | 0.622 | 0.610 | 0.646 | 0.667 | 0.773 | 0.518 | 0.605 |
| | XGBoost | **0.741** | 0.622 | 0.617 | 0.593 | 0.668 | 0.693 | 0.795 | 0.561 | 0.610 |
| | CatBoost | 0.721 | 0.610 | **0.629** | 0.600 | 0.677 | 0.701 | 0.801 | 0.558 | **0.620** |
| | MLP | 0.701 | 0.621 | 0.625 | 0.597 | 0.668 | 0.687 | 0.763 | 0.558 | 0.553 |
| | RealMLP | 0.730 | **0.630** | **0.629** | **0.617** | 0.654 | 0.675 | 0.793 | 0.546 | 0.615 |
| | ModernNCA | 0.732 | 0.622 | 0.622 | 0.611 | 0.655 | 0.679 | 0.796 | 0.540 | 0.608 |
| | TabPFN v2.5 | 0.584 | 0.500 | 0.500 | 0.500 | **0.796** | **0.760** | **0.806** | 0.567 | 0.619 |

**TabPFN v2.5 and tree-based models do not compete along the same dimension, while they represent two fundamentally distinct paradigms of intelligence.** As evidenced by the average ranks in Table 1, this distinction manifests in their respective strengths. Tree-based models, particularly CatBoost, excel in scenarios involving feature shifts and unstable input schemas, challenges such as decremental/incremental features and changing data distributions, drawing on a feature-driven, locally partitioning form of intelligence that enables robust survival adaptability. In contrast, TabPFN v2.5 demonstrates superior performance in tasks requiring semantic-level adaptation, achieving the best rank in emerging new classes with statistically significant superiority over all six baselines and varied learning objectives, driven by its prior-driven, globally inferential, and uncertainty-aware intelligence that supports high-level decision transfer.

**Insights and implications.** Beyond the overall rankings, our study reveals both expected and non-trivial behaviors of TabPFN v2.5. As anticipated, its reliance on a fixed input dimensionality prevents it from handling newly added features at test time, and its performance degrades substantially under concept shift while showing greater resilience to covariate shift. However, several findings are more surprising. TabPFN v2.5 demonstrates a strong capability in emerging new class detection, achieving statistically significant improvements over most baseline methods, yet exhibits heightened vulnerability to decremental features, particularly in regression tasks, where its performance gap widens considerably as feature shift increases. Additionally, given the fundamental differences in applicable scenarios between TabPFN v2.5 and tree-based models, model selection should be aligned with the specific capabilities prioritized by the target scenario in open environments.

### F.6. Recommendations

During the experimental investigation, we observed that the majority of existing high-performance models predominantly demonstrate their superior performance in closed environments. However, these models tend to fall short in adapting to the open environments challenges that are more frequently encountered in real-world scenarios. To further enhance the performance of models in open environments and to provide guidance for the development of subsequent research, the following recommendations are proposed:

**Develop comprehensive benchmarks and evaluation protocols for open environments tabular learning.** Existing

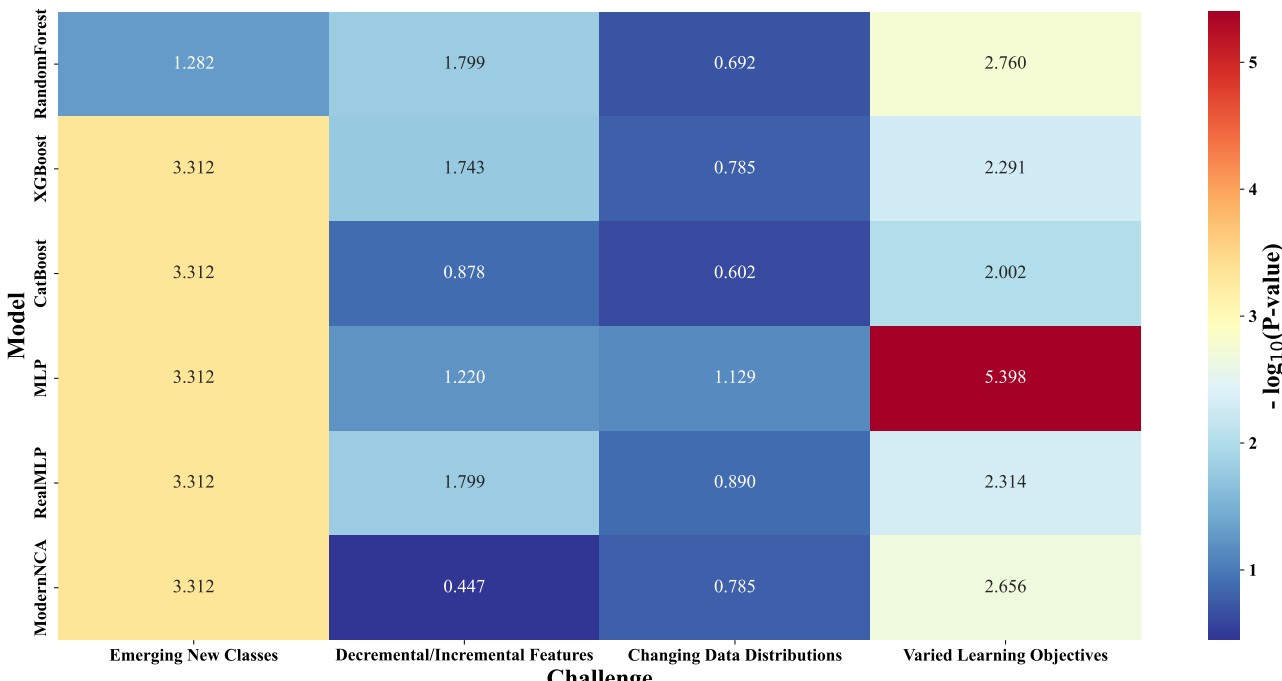

*Figure 5.* $Log_{10}$(P-value) of six models compared with TabPFN v2.5 across four challenge dimensions. Higher values indicate stronger evidence of superiority, with values exceeding 1.30 denoting statistical significance.

benchmarks predominantly address distribution and feature shifts, neglecting critical aspects like emerging classes and dynamic learning objectives. Current evaluation metrics (e.g., OOD accuracy, performance gap) also remain confined to traditional distribution shifts, overlooking broader open environments challenges. To address these limitations, new benchmarks should integrate diverse tasks in open environments, while evaluations should adopt specialized metrics (e.g., Open-World Tracking Accuracy (Liu et al., 2022), mean average precision (Sancaktar et al., 2022)) to rigorously assess adaptation to novel classes, task shifts, and evolving objectives. Together, these advances will enable more precise performance measurement and drive methodological progress in tabular learning under open environments settings.

**Take model robustness as a critical metric when comparing model quality.** Current research often judges the model quality solely based on its performance in closed environments, without considering its robustness in open environments as an important evaluation criterion. However, robustness is a crucial indicator of a model's practical value, and excellent performance in closed environments does not necessarily mean that the model will perform well in open environments. Therefore, robustness should be regarded as a critical metric when comparing model quality, and the quality of a model should be comprehensively assessed based on both closed environments performance and open environments robustness.

**Design specific modular architecture and adaptive mechanisms for TabPFN v2.5.** Future architectural improvements to TabPFN can be pursued along several directions. First, to address the inherent limitation of fixed input dimensionality, dynamic feature adaptation mechanisms can be developed, such as extensible feature encoders or attention-based modules, so that the model can accommodate variations in the feature set at test time. Second, to mitigate its vulnerability under concept drift, online adaptation modules can be introduced to enable dynamic adjustment of the mapping rules. Furthermore, multi-objective prediction heads with gating mechanisms can be integrated to adapt a single backbone network to diverse evaluation metrics, while scale-aware routing modules can be incorporated to preserve its strength in small-sample settings while extending applicability to large-scale data.

**Simulate open environments in the pre-training phase via large-scale synthetic data generation.** Drawing on the successful paradigm of training TabPFN on synthetic data generated from structural causal models, a systematic synthetic benchmark generation framework can be developed. By defining a set of configurable change operators, including class-level, feature-level, distribution-level, and objective-level variations, such a framework enables controlled simulation of the combinations and intensities of various open-environment challenges. This approach not only exposes the model to open-environment conditions during training but also establishes a standardized evaluation benchmark that facilitates fair model comparison and decoupled analysis of adaptability.

