# OpenReview forum: "Realistic Evaluation of TabPFN v2.5 in Open Environments"
_ICML.cc/2026/Workshop/FMSD — FMSD @ ICML 2026 Poster_

### Official Review · Reviewer_MxNy · 2026-05-21
**Review of the paper: "Realistic Evaluation of TabPFN v2.5 in Open Environments"**

**Rating:** 7
**Confidence:** 4

**Review:**

**Strengths:**

- The paper addresses a genuinely underexplored gap by being the first to evaluate TabPFN v2.5 across four open-environment dimensions rather than a single shift type. I personally also enjoy seeing the paper discussing the failure modes of tabular foundation models (TFMs) while most paper only discuss the advantages.
- The evaluation is broad. The paper integrates three existing benchmarks (TabFSBench, WhyShift, TableShift) into one pipeline, reports multiple metrics per challenge, applies Wilcoxon tests, and compares against a diverse baseline set of tree-based and deep learning models.
- The paper closes with four concrete, observation-linked recommendations naming specific architectural directions (dynamic feature adaptation, online adaptation, multi-objective heads, scale-aware routing) and a synthetic-data strategy for open-environment pretraining.
- It’s great to have the anonymous code repository for reproducibility.
- I personally like the writing style of this paper by making the conclusion of each paragraph in bold. This make it really easy for the reviewers.

**Weakness, Questions, and Suggestions:**

- **Statistically insignificant (?):** The results is from “the average of three different random seeds“ (Section E.1). This is statistically underwhelming for any machine learning (ML) results. Most commonly people at least do 10 runs/seeds to claim statistical significant in the results. It would be great to see a more rigorous evaluation on this in the camera ready version. Additionally, for all results in this paper, what are the Wilcoxon tests setup (paired across datasets? metrics? seeds?)
- **Table 5’s results:** Table 5 (regression) shows MLP starts at RMSE 0.997 at 0% shift while TabPFN v2.5 starts at 0.803 (lower is better). At 100% shift, MLP reaches 1.247 (gap +0.250) and TabPFN v2.5 reaches 1.723 (gap +0.920). The paper concludes from this gap comparison that MLP is "more robust." My concern was that the gap-based metric can be misleading when models start at very different baselines, because a model that doesn't strongly exploit features in the first place will seems robust under feature removal. Maybe some kind of normalization is required here? I wonder if there are literature that justify this metric/interpretation, or a more robust mathematical or statistical justification.
- **Why 3 uncertainty intervals?:** Can the authors justify the choices of the three uncertainty intervals, [0.4, 0.6], [0.45, 0.55], and [0.49, 0.51], in Table 4's results? What counts as “uncertain” here? Are there literature or interval formulation that can support this choice?
- **Questions on F.2. Decremental/Incremental Features:** I believe clarifications on the experiment details is needed. Are multiple random feature sampled at each shift level and results averaged, or whether one subset is drawn per level? Are the *same* removed features used across all seven models at each level (essential for a fair comparisons)? Is feature importance taken into account (random removal treats highly informative and uninformative features identically, which is one defensible choice but should be stated)? Are the features include both categorical and numerical values, and whether categorical and numerical features are removed proportionally or by uniform random draw.
- **Questions on F.3. Changing Data Distributions’ results:** 1. In the results in Table 6, it is unclear to me what columns are covariate shift problems and what are concept shift problems. Adding a label would help. 2. The results only show the accuracy comparison of on TabPFN v2.5 vs. XGBoost on changing data distributions tasks. What about the Wilcoxon analysis and rank? If the claim is that “TabPFN v2.5 appears to handle covariate shift better than concept shift,” shouldn’t there be a statistical significant (rank) across all models for covariate shift and concept shift problems respectively to support this statement rather than only averaged accuracy?
- **Hyperparameter optimization (HPO) between the baselines and TabPFN v2.5 :** The baselines undergo extensive hyperparameter search (App. E.3), while TabPFN v2.5 appears in no HPO and is evaluated only in its default "vanilla" configuration. However, TabPFN has tunable preprocessing, ensembling, and post-hoc options exposed through the tabpfn-extensions library (e.g., AutoTabPFN, post-hoc ensembling), which the TabPFN-2.5 report and TabArena paper both treat as settings. The established protocol in TabArena is to benchmark every model (foundation models included) in three regimes: default, tuned, and tuned+ensembled, then perform apple-to-apple comparisons. I am wondering if the evaluation protocol in this paper would result in somewhat ‘unfair’ comparison between the tree-based/deep-learning models vs. the vanilla TabPFN. The authors need to justify their comparison protocol.

Overall I think it’s a nice workshop paper. However, for a full paper, the authors would have to resolve the questions in the evaluation protocol and result presentation above.

---

### Official Review · Reviewer_ZGtC · 2026-05-22
**Review: Realistic Evaluation of TabPFN v2.5 in Open Environments**

**Rating:** 6
**Confidence:** 4

**Review:**

## Summary
The paper evaluates TabPFN v2.5 under “open environment” tabular-learning challenges: emerging new classes, feature shifts, distribution shifts, and varied learning objectives. It proposes a unified evaluation framework and compares TabPFN v2.5 against tree-based and deep-learning baselines, concluding that TabPFN v2.5 is strong for uncertainty-driven novel-class detection and multi-metric performance, but weaker under decremental features and concept shift.

## Strengths
- Strong fit for the workshop.
- Addresses the underexplored question of whether TFMs remain robust outside simple i.i.d. settings.
- The recommendations for future benchmarks, robustness metrics, and adaptive TabPFN modules are useful.

## Areas for Improvement
- Most of the open environment challenges can as well be seen as problems of pipeline design and not of foundation models in particular, as they concern every model. A better explanation of why a model should even be able to handle problems like varied objectives natively would help. Currently several of the problems can be resolved as part of the problem design.
- Varied learning objectives are less clearly motivated as an open-environment challenge. The paper should better explain realistic use cases where the objective changes after deployment. I find it hard to believe that practitioners don’t already know and evaluate on what matters before deployment.
- The emergence of new classes may be less universal in tabular data than in image or text domains, so the paper would benefit from more concrete tabular use cases. Also, the emergence of new classes changes the problem definition and means that retraining is necessary once enough new class examples are collected.
- How is the problem of decremental features different from missing values? The only difference is that for decremental features the issue is clearly systematic. However, it should be known upfront whether this is possible and can be fixed during the task definition. This seems at least partly a pipeline-design issue, and the paper should clarify why model-level robustness to decremental features is a necessary focus.
- More details on dataset selection, preprocessing, and splits would strengthen the paper. In the current version the reproducibility from the paper alone is limited. Also, clarification on how datasets are selected for each challenge and whether conclusions are sensitive to dataset choice would be helpful.
- The novelty of the unified framework over combining existing benchmarks could be clarified.
- Some claims at the end of the paper are hard to understand and the formulation is unusual. For example, the formulation “prior-driven decision model rather than a purely data-fitting tabular predictor” is odd, since “decision model” and “tabular predictor” could be used interchangeably. Also, “TabPFN v2.5 and tree-based models […] represent two fundamentally distinct paradigms of intelligence” is a strong claim and would require more explanation and experiments.
- Only one tabular foundation model is studied, which limits how far the conclusions generalize to TFMs as a class.

---

### Official Review · Reviewer_mP7z · 2026-05-22
**Realistic Evaluation of TabPFNv2.5 in open environment settings**

**Rating:** 7
**Confidence:** 4

**Review:**

# Summary

The paper presents the first systematic evaluation of TabPFN v2.5 under open-environment conditions across four challenge categories: emerging new classes, decremental/incremental features, changing data distributions, and varied learning objectives. The authors have discovered that TabPFN v2.5 behaves as a prior-driven decision model that excels at semantic-level adaptations (new class detection, varied objectives, class imbalance), but doesn’t perform as well with perturbations in data (decremented features, concept shift). They also propose a unified evaluation framework integrating TabFSBench, WhyShift, and TableShift.

# Strengths

- The problem is well identified and defined. There does exist a gap between closed environment benchmarking and real-world deployment, and the authors have correctly chosen to address this with evaluating the TabPFN model.
- I found the framing of TabPFN’s strengths and weaknesses as TabPFN being a prior-driven global inference engine rather than a conventional data-fitting model to be quite useful! Certainly more useful than standard leaderboard based benchmarking, as it helps understand the utility of the TabPFN model better.

# Areas of Improvement

- Only using 4 datasets for emerging clew class analysis seems a little low. I would have liked a more comprehensive analysis here.
- The p<0.0001 claim for emerging classes excludes Random Forest, a model used frequently for tabular datasets. This should be made upfront rather than mentioning in the appendix.
- I would have liked the authors explaining some more about why regression performance specifically is reduced under decremented features. This is an interesting observation, but warrants some inherent mathematical explanation. MLP’s “inherent anti-shift” properties doesn’t explain much!
- TabPFN v2.5’s accuracy ranging from 0.594 to 0.944 within the covariate shift category alone suggests the categorization may be too noisy to support the conclusions drawn.

The authors have identified a good problem to solve with evaluating TabPFN in real world open settings rather than closed benchmarks. I recommend the paper be accepted with revisions, with the authors addressing the regression performance and performance variance in the covariance shift category.